# Evaluation of radiological capacity and usage in paediatric TB diagnosis: A mixed-method protocol of a comparative study in Mozambique, South Africa and Spain

Isabelle Munyangaju[1,2,3], Benedita José[4], Ridwaan Esmail[5], Megan Palmer[6], Begoña Santiago[7], Alicia Hernanz-Lobo[7,8,9], Crimenia Mutemba[4], Patricia Perez[10], Liebe Hendrietta Tlhapi[11], Vanessa Mudaly[12], Richard D. Pitcher[13], Andreas Jahnen[14], Eliseo Vañó Carruana[15], Elisa López-Varela[1,2], Isabelle Thierry-Chef[1,2,3,16]*

1 Barcelona Institute for Global Health, Barcelona, Catalonia, Spain, 2 Medicine and Translational Research Department, University of Barcelona, Barcelona, Spain, 3 CIBER Epidemiología y Salud Pública (CIBERESP), Madrid, Spain, 4 National Tuberculosis Control Program, Ministry of Health, Maputo, Mozambique, 5 Imaging Division, Ministry of Health, Maputo, Mozambique, 6 Desmond Tutu TB Centre, Department of Paediatrics and Child Health, Faculty of Medicine and Health Sciences, Stellenbosch University, Cape Town, South Africa, 7 Pediatric Infectious Diseases Department, Gregorio Marañón University Hospital, Gregorio Marañón Research Health Institute (IiSGM), Madrid, Spain, 8 Centro de Investigación Biomédica en Red de Enfermedades Infecciosas (CIBERINFEC), Instituto de Salud Carlos III, Madrid, Spain, 9 Translational Research Network in Pediatric Infectious Diseases (RITIP), Madrid, Spain, 10 National Paediatric TB Working Group, Maputo, Mozambique, 11 Radiation Control, South African Health Products Regulatory Authority, South Africa, 12 Service Priorities Coordination (SPC) Directorate, Department of Health, Western Cape, South Africa, 13 Department of Medical Imaging and Clinical Oncology, Faculty of Medicine and Health Sciences, Stellenbosch University, Cape Town, South Africa, 14 ITIS Department, Luxembourg Institute of Science and Technology, Luxembourg, 15 Department of Radiology, Faculty of Medicine of the Complutense University, Madrid, Spain, 16 Universitat Pompeu Fabra (UPF), Barcelona, Spain

☯ These authors contributed equally to this work.

* isabelle.thierrychef@isglobal.org

**Data Availability Statement:** No datasets were generated or analysed during the current study.

## Abstract

### Introduction

Tuberculosis remains one of the top ten causes of mortality globally. Children accounted for 12% of all TB cases and 18% of all TB deaths in 2022. Paediatric TB is difficult to diagnose with conventional laboratory tests, and chest radiographs remain crucial. However, in low- and middle-income countries with high TB burden, the capacity for radiological diagnosis of paediatric TB is rarely documented and data on the associated radiation exposure limited.

### Methods

A multicentre, mixed-methods study is proposed in three countries, Mozambique, South Africa and Spain. At the national level, official registry databases will be utilised to retrospectively compile an inventory of licensed imaging resources (mainly X-ray and Computed Tomography (CT) scan equipment) for the year 2021. At the selected health facility level, three descriptive cross-sectional standardised surveys will be conducted to assess radiology capacity, radiological imaging diagnostic use for paediatric TB diagnosis, and radiation

Deidentified research data from this study will be made publicly available when the study is completed and published.

**Funding:** We acknowledge support from the grant CEX2018-000806-S funded by MCIN/AEI/ 10. 13039/501100011033, and support from the Generalitat de Catalunya through the CERCA Program. Isabelle Munyangaju the support of a fellowship from "la Caixa" Foundation (ID 100010434). The fellowship code is LCF/BQ/DI21/ 11860045. Role of the funding sources: None of the funding sources had a role in the study's design, conduct and reporting.

**Competing interests:** The authors have declared that no competing interests exist.

protection optimization: a site survey, a clinician-targeted survey, and a radiology staff-targeted survey, respectively. At the patient level, potential dose optimisation will be assessed for children under 16 years of age who were diagnosed and treated for TB in selected sites in each country. For this component, a retrospective analysis of dosimetry will be performed on TB and radiology data routinely collected at the respective sites. National inventory data will be presented as the number of units per million people by modality, region and country. Descriptive analyses will be conducted on survey data, including the demographic, clinical and programmatic characteristics of children treated for TB who had imaging examinations (chest X-ray (CXR) and/or CT scan). Dose exposure analysis will be performed by children's age, gender and disease spectrum.

## Discussion

As far as we know, this is the first multicentre and multi-national study to compare radiological capacity, radiation protection optimization and practices between high and low TB burden settings in the context of childhood TB management. The planned comparative analyses will inform policy-makers of existing radiological capacity and deficiencies, allowing better resource prioritisation. It will inform clinicians and radiologists on best practices and means to optimise the use of radiological technology in paediatric TB management.

## Introduction

Tuberculosis (TB) remains one of the top-ten causes of mortality globally with an estimated 10.6 million cases in 2022 and 1.3 million deaths [1]. Sub-Saharan Africa bears a significant share of the global burden of HIV/AIDS and TB, these conditions being mutually reinforcing [1, 2]. The COVID-19 pandemic and associated social restrictions negatively impacted the TB epidemic, reversing the progress made in case detection and mortality statistics [1, 3, 4].

Childhood TB is historically difficult to diagnose and the spectrum of disease is wide, particularly in the setting of HIV co-infection and when compared to adults [5, 6]. The disease is paucibacillary and there can be difficulty in obtaining quality samples with high yield. Microbiological confirmation in children is achieved in <10–15% of cases for sputum microscopy, 5–45% for GeneXpert molecular tests and < 30–40% for culture [7, 8]. HIV alters the pathogenesis of TB and TB-specific adaptive immunity, which can lead to atypical presentations of TB disease [9]. Furthermore, TB-HIV co-infected children are susceptible to other opportunistic respiratory infections with overlapping clinical presentations which can compound the diagnostic challenge [9, 10]. Due to these diagnostic challenges, imaging studies remain important as paediatric TB continues to often be diagnosed without microbiological confirmation, using clinical algorithms or accepted combinations of standard references that include a chest X-ray (CXR) [or Computed Tomography (CT) scan] [6, 11].

The latest World Health Organization (WHO) paediatric guidelines strongly recommend the inclusion of chest radiography both in diagnostic algorithms and for the evaluation of disease severity. In these new guidelines, CXR findings are crucial in determining whether a child diagnosed is eligible for a 4-month treatment regimen, for non-severe disease, or a 6-month regimen for severe disease. The CXR can also assist in excluding TB in children eligible for TB preventive therapy (contact tracing), excluding alternative respiratory diagnoses and evaluating responses to TB treatment [12].

The use of a CT scan for the diagnosis of TB is increasing, particularly in high-income countries, due to its superior diagnostic accuracy compared to the CXR. One study demonstrated that CT scanning was able to detect TB in 70% of the cases compared to 37% for CXR [13]. CT scans are regarded as the gold standard for detecting lymphadenopathy in childhood TB. CT scans allow for a clear assessment and staging of complications of lymphadenopathy such as airway obstruction [14].

There are limitations to the use of imaging in the diagnosis and management of TB. These include the high cost of equipment acquisition and maintenance, the need for expert human resources to interpret findings, low sensitivity and specificity (particularly for CXR), inter-reader and technical variability, as well as the risks of ionising radiation, particularly for CT scans [15–18]. For children requiring repeated testing, such as those living with HIV who are at risk of recurrent TB [19] and who usually need more than one diagnostic procedure, radiation exposure is an important factor to consider and optimise.

Since the new WHO recommendations propose a shortened treatment regimen for non-severe disease, national TB programs may consider expanding CXR usage to implement these guidelines. The first step in implementation is an evaluation of a country's radiological capacity. The WHO guidelines state that one X-ray and one ultrasound unit for every 50,000 people can meet 90% of all imaging needs in low- and middle-income countries (LMICs) [20]. There are limited published data on national radiological capacity in low- and middle-income countries with a high TB burden. About seven studies have been carried out in South Africa, Tanzania, Zimbabwe, Zambia, Uganda, Ghana and Kenya mapping out their radiological capacity and its geographical distribution [21–28]. In Zambia, the study found that the number of rural imaging facilities was much higher than that of urban areas but more than 80% of the human resources required for radiology were located in urban areas. There were only 5.7 general radiography units per million people in Tanzania's public sector, well below the WHO's recommendation of 20 units per million [20, 29].

This multicentre and multidisciplinary study aims to assess the radiological capacity (in terms of radiological equipment, human resources and radiological protocols) and to evaluate the cumulative ionising radiation dose exposure for the radiological diagnosis and management of childhood TB in three different economic settings, namely Mozambique (low income, high TB and HIV burden setting), South Africa (middle income, high TB and HIV burden setting) and Spain (high income, low TB and HIV burden) (see Table 1).

### Study objectives

Table 1.

## Materials and methods

The study objectives will be achieved through several approaches (see Fig 1). Currently, the study is in the data collection stage.

In each country, we selected a sample of hospitals and clinics based on:

- TB burden–The selected health facilities were those with the highest number of children treated for TB over the past five years. This means we can recruit a substantial number of children for the dosimetry study.

- TB services access–Our focus was on facilities where the TB disease was managed (TB clinics or infectious disease units) and where TB registers or databases existed.

- Radiology access–A radiology department is located in at least one of the participating health facilities and other health facilities have referral access to radiological imaging (CXR and/or

**Table 1. The study-specific objectives.**

|  | Objectives | Outcomes |
|---|---|---|
| **Specific objective 1** | **At the national level:** evaluate the radiological capacity for childhood TB diagnosis and management in Mozambique, South Africa and Spain by mapping the number of CXR units per million population and the number of CT scan units per million population. | 1) Quantitative mapping of CXR and CT scan units in all three countries<br>2) Geographical mapping of the distribution of CXR and CT scan units within each country, highlighting regional variations<br>3) Radiological capacity variations across different population densities and TB disease burden between countries and regions.<br>4) Insights into potential infrastructure and resource differences between Mozambique, South Africa, and Spain |
| **Specific objective 2** | **At the health facility level:** evaluate the radiological human resources a, availability and utilisation of typical radiological protocols, clinical and radiology staff experience and knowledge on the use of radiological diagnostics for childhood TB diagnosis in Mozambique, South Africa and Spain. | 1) Identification of the number and distribution of radiological staff (e.g., radiologists, radiographers, technicians) in each country<br>2) Evaluation of the training and qualifications of radiological personnel involved in childhood TB diagnosis.<br>3) Documentation of the typical radiological protocols used for childhood TB diagnosis in healthcare facilities in Mozambique, South Africa, and Spain<br>4) Identification of challenges or barriers faced by healthcare professionals in the use of radiological diagnostics for childhood TB |
| **Specific objective 3** | **At the patient level:** evaluate the exposure patterns by age groups and by disease spectrum ; further investigate the potential for exposure reduction through dose optimisation keeping in mind the benefit-risk balance. | 1) Quantitative data on ionizing radiation exposure for different age groups<br>2) Identification of age-specific trends or variations in radiation exposure levels<br>3) Stratified analysis of ionizing radiation exposure, providing insights into how exposure patterns vary across specific age ranges<br>4) Recognition of demographic groups or patient populations that may be at higher risk of ionizing radiation exposure, considering both age and disease spectrum |

[a] The study will only evaluate facility human resources instead of national-level health resources, as 1) the primary endpoint of the evaluation is to understand the exposure patterns in children with TB and explore the potential for dose optimisation, 2) the human resources evaluation is to complement the findings of the primary endpoint and 3) there are feasibility considerations.

CT scan). We selected health facilities that have a perceived functioning radiological diagnostics network since our primary endpoint is to evaluate exposure patterns of children with TB.

• Feasibility–Due to financial constraints, we selected health facilities that were within reach of partnering institutions in each country.

## 1. Methodology for mapping radiology capacity

**At the national level** in all three countries, we will perform a retrospective audit of licensed X-ray and CT units for the year 2021 using the official registries of the Mozambican National Imaging Division, the South African Health Products Regulatory Authority (*SAHPRA*) and the Spanish Nuclear Security Council (*CSN–Consejo de Seguridad Nuclear*).

  **At the selected health facility level** in all three countries, the radiological capacity and radiation protection protection will be assessed through two standardised online surveys: 1) a site survey targeting site operational managers (1 operational manager per site) and conducted by

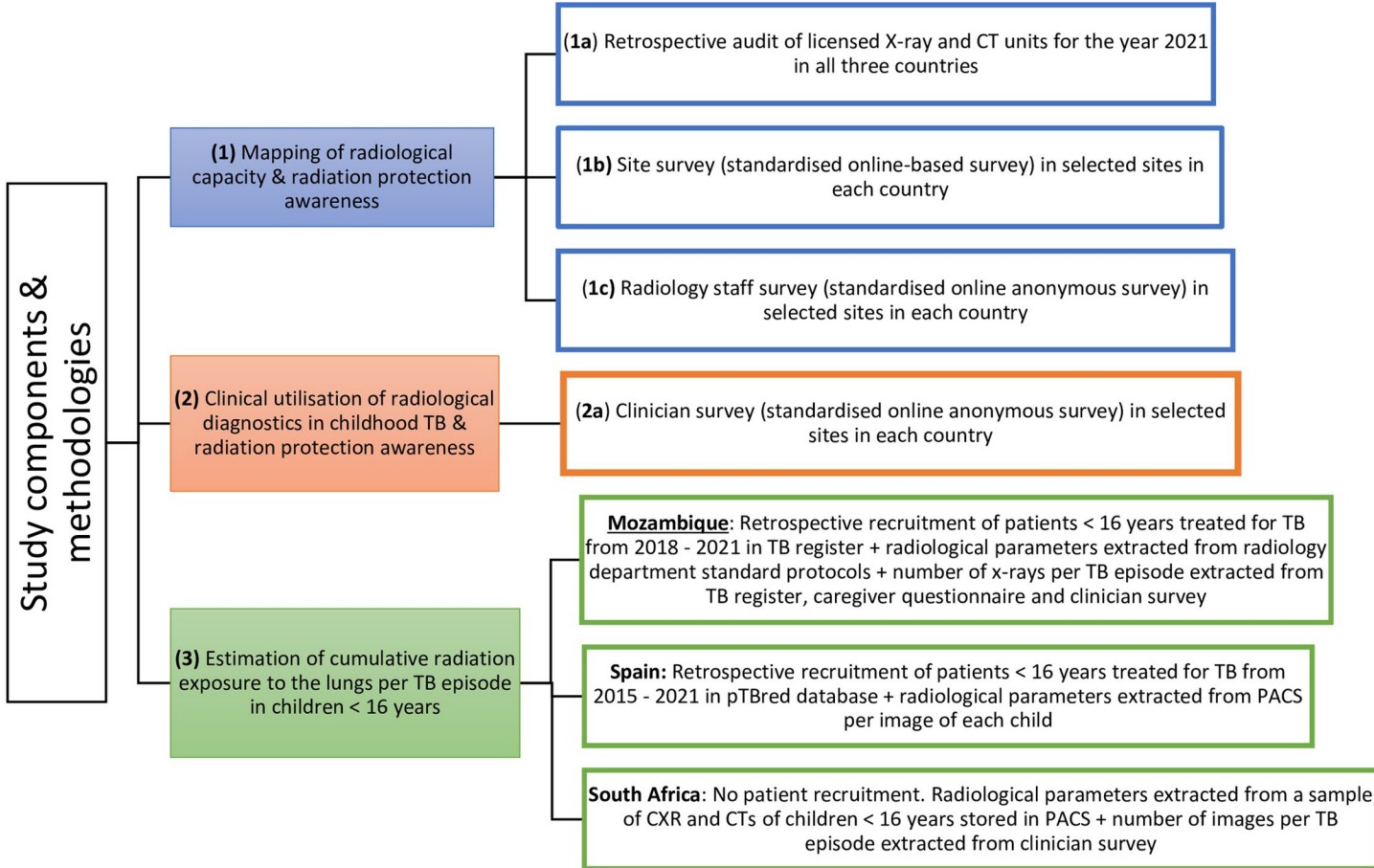

**Fig 1. Study components and methodologies.** In the figure, we describe the different study methodologies applied to answer each study objective (CT: computed tomography scan; TB: tuberculosis; PACS: Picture Archiving and Communications System; CXR: chest x-ray).

a study data collector to acquire data on existing on-site radiological equipment, clinical and radiological human resources, radiological protocols available and radiological processes, and 2) a standardised anonymous online survey targeting all radiology staff (radiologists, radiographers, trainee radiologists and trainee radiographers) involved in the diagnostic imaging of children with presumed TB. This survey will gather data on the use of typical radiological protocols and radiation protection optimization by radiology staff involved in TB care. Participants will need to give informed consent to participate in this survey. To complete the national data that are collected per province/region and not per facility, we will also collect data on available radiological equipment per selected site. The radiology personnel survey will be conducted on the Google Forms platform and the survey link will be sent to the targeted population individually via email and/or phone as provided by each selected site's operational manager. Participants will be given two months to answer the survey, with reminders sent every two weeks (a total of three reminders) to stimulate participation.

## 2. Methodology for evaluating clinical utilisation of imaging in paediatric TB and radiation protection optimization

A third standardised online anonymous survey will be administered to all clinical personnel involved in the screening, diagnosis and management of children with presumed TB in each

country site. Participants will be identified with the assistance of the health facility operational manager. The manager will enlist and keep a log of all clinical staff that fit the aforementioned criteria and share the survey links via email or phone. This survey will also be conducted online using Google Forms and the link will be sent to the targeted population individually via email and/or phone. Participants will need to give informed consent to participate in the survey. Participants will be given two months to answer the survey, with reminders sent every two weeks (a total of three reminders) to stimulate participation.

The survey will collect data on the clinical use of radiological diagnostics and radiation protection optimization by clinicians involved in childhood TB management. In Mozambique, the targeted population includes general practitioners, medical assistants (or medical technicians) and prescribing nurses. In South Africa, the clinicians will include general practitioners, family medicine practitioners and paediatricians. In Spain, paediatricians in the paediatric infectious diseases departments of the selected hospitals as well as those registered in the Spanish Paediatric TB Research Network (pTBred) will be included.

To maintain comparability across the three countries, the study does not include medical physicists in the surveys. The sites selected in Mozambique have no medical physicists, and there are few in the country. However, where possible, we had the support of medical physicists for the collection of radiological parameters and radiological protocols.

## 3. Methodology for estimating cumulative radiation dose exposure to the lungs per TB episode in children < 16 years

To perform the dosimetry (estimate the dose exposure pattern) we will adapt to each country's radiological capacity and local privacy laws (see Fig 2). In Mozambique, where there is no Picture Archiving and Communications System (PACS), dosimetry will be calculated using the individual clinical data of children under 16 years registered in the TB register from 2018 to

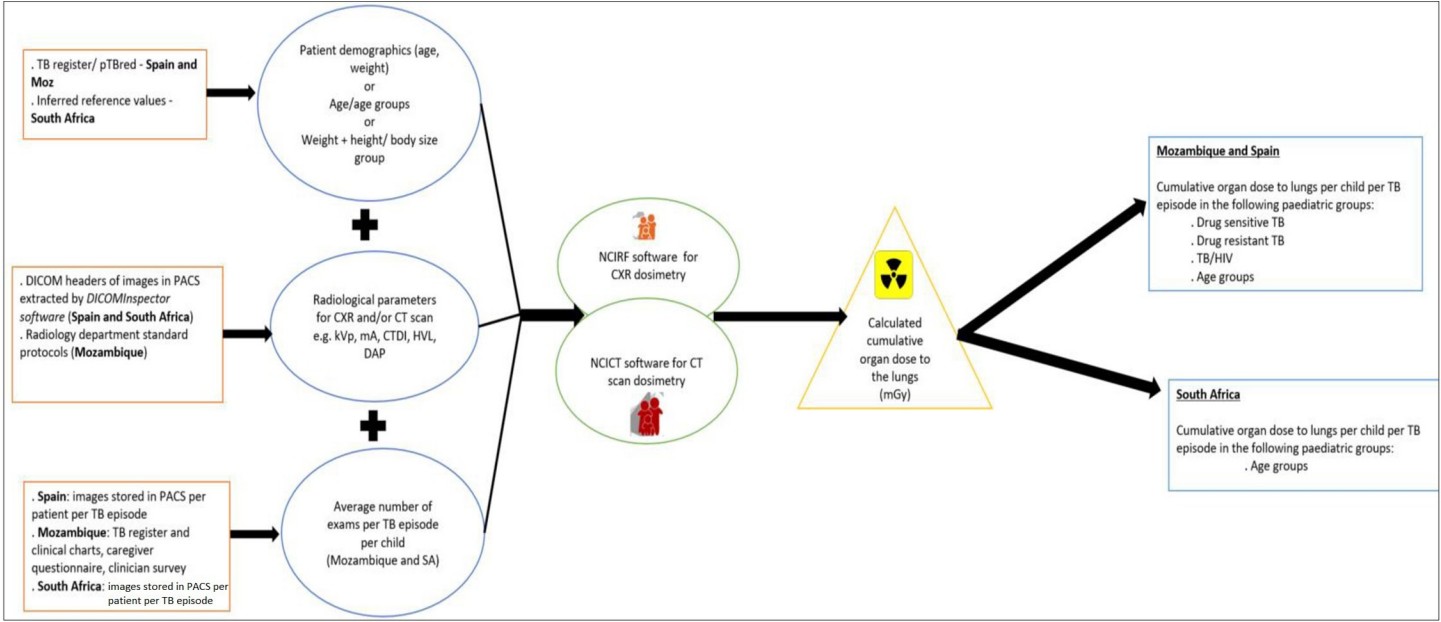

**Fig 2. Dosimetry chart.** In this figure, we describe the procedures for obtaining the dosimetry data in each country and the calculation of the cumulative organ dose to the lungs (PACS: Picture Archiving and Communications System; TB: tuberculosis; CXR: chest x-ray; CT: computed tomography scan; kVp: kilovoltage peak; mA: milliamperes; CTDI: computed tomography dose index; HVL: half-value layer; DAP: dose area product; SA: South Africa; NCIRF: National Cancer Institute dosimetry system for Radiography and Fluoroscopy; NCICT: National Cancer Institute dosimetry system for Computed Tomography; mGy: milligray).

2021, radiological parameters extracted from the standard protocols used by the radiological departments of selected sites, and the number of CXR per child during the TB episode will be extrapolated from the TB register, guardian/caregiver questionnaire (consent form for the guardian and assent form for adolescent will be provided) and clinician survey (consent form will be provided).

In South Africa, this methodology will be adapted to align with local regulations and use the most accurate methodology (providing data which will be more comparable with those obtained in the other 2 countries). Radiological parameters for each TB case treated in year 2022 will be extracted, for each clinical record, by the site PACS manager. The associated clinical data will be extracted by site clinicians and each patient record will have a unique study number which will be provided to the study team. As a result, the study team will not be able to access patient identifiers, as required by the Protection of Personal Information Act.

In Spain, where there is a PACS, dosimetry will be calculated from individual clinical data on children under 16 years registered in the pTBred database from 2015 to 2021 (this is due to the low number of paediatric TB cases seen in Spain) and radiological parameters as extracted from the Digital Imaging and Communications in Medicine (DICOM) headers of each of the images of each individual child. In both cases, the variables include demographic, clinical, radiological and machine specification data to assess the exposure pattern per child.

For images obtained from the PACS in Spain and South Africa, the radiological data will be extracted from the DICOM tags of the images using the software DICOMInspector which uses the local patient ID to trace the images in PACS and records machine settings in a pseudonymised format. Using the radiological data, demographic data and number of exams typically performed per TB episode, the cumulative radiation exposure will be calculated per individual child treated for TB using the dedicated software tools of the National Cancer Institute (NCIRF *[National Cancer Institute dosimetry system for Radiography and Fluoroscopy]* and NCICT [*National Cancer Institute dosimetry system for Computed Tomography]*). These computational tools are used to estimate the organ doses for children and adults undergoing X-ray and CT scan examinations [30–32].

## Data analysis

At the national level X-rays and CT scans will be reflected as the number of machines per million people for each region and for the country as a whole, based on the available and most updated population estimates for each country. A comparative assessment of the capacity between the three countries, as well as the available international standards (WHO and Organisation for Economic Co-operation and Development (OECD)), will be performed.

Survey data will be analysed using descriptive statistics. The surveys will enhance the national mapping and dosimetry findings. With regard to the national mapping analysis, the surveys will provide a snapshot of the radiological capabilities and function at the health facility level. The surveys will complement the dosimetric findings with an analysis of human resource capabilities, the use of image diagnostics (including frequency of radiology exams during a TB episode), and an awareness of radiation protection in paediatric TB management.

For the dose exposure analysis, children will be described by age, gender and disease spectrum. Exposure dose estimation will be conducted using National Cancer Institute tools; the units for dosimetric analysis will be absorbed dose to the lungs, heart and breast (milligray = mGy). The Chi-test, Fisher's exact test and correlations will be conducted to evaluate the relationship between categorical variables (e.g. if TB/HIV co-infected children have a higher frequency of exposure than the children who are not co-infected). Descriptive analyses

will be conducted to summarise participant demographic and clinical characteristics (age, HIV status, new TB case, pulmonary or extra-pulmonary, amongst others).

## Study sample

The overarching aim of the study is to investigate ways to optimise radiological imaging use in the diagnosis of paediatric TB, not to evaluate the risk associated with exposure. For this reason, no sample size calculation was estimated. We propose looking at all children under 16 years diagnosed and treated for TB in the selected sites for the study duration in Mozambique and Spain. These children will be included from the TB register in Mozambique and the pTBred in Spain. We expect to include data from 350 children in Mozambique and 200 in Spain. In South Africa no patient data will be included.

For feasibility reasons, we selected no more than five facilities in each country based on the previously described selection criteria. We chose to make the clinician and radiology staff survey available to all staff involved in screening, diagnosing, treating and managing paediatric TB in the clinical and radiology departments of each health facility. We took into account the average response rate for an online survey (typically 40%) [33] and the differences between countries in human resource capacity.

## Data management

For the national level radiology capacity mapping, data will be captured on a customised spreadsheet and stratified by imaging modality, installation (fixed/mobile), healthcare sector (public/private) and geographic region. These data will then be uploaded into a dedicated study REDCap® database. For the site level radiology capacity mapping, the site survey will be performed directly into the study-specific REDCap® database. The radiology personnel survey and the clinician survey will be conducted on the Google Forms platform and responses will be aggregated in the study's dedicated database.

For the estimation of cumulative radiation dose exposure: clinical, radiological and dosimetry data will be compiled in the study's dedicated REDCap® database. To ensure confidentiality and safety, all patient information (where collected) will be captured using a study-specific identification number and the list linking this number to the patient's personal details will only be accessible (under informed consent) to the principal investigator or another person as delegated by them. Data will be kept for as long as it is necessary for the purpose of the research project and according to applicable laws or up to a maximum of five years after the last publication to allow for potential re-analyses. After these five years, the data will be destroyed or sent back to the participating centres, at the discretion of each.

## Ethical considerations and dissemination

This study and the informed consents have been elaborated in strict compliance with the ethics principles of the 1964 Declaration of Helsinki and its later amendments regarding human participants and their data or biological material. Ethical approval for this study (including approval of the informed consent and assent forms) was obtained from the following ethics committees: *Comité de Ética de la Investigación con medicamentos del Parc de Salut MAR—Spain* (CEImPSMAR Ref: 2022/10358); *Comité Nacional de Bioética para a Saúde–Mozambique* (CNBS Ref: 614/CNBS/22) and *Health Research Ethics Committee–South Africa* (HREC Ref: N22/08/105). The findings from this study will be disseminated locally and internationally through manuscript publications in peer-reviewed journals and conference presentations over national and international platforms.

## Discussion

To the best of our knowledge, this is the first multicentre and multi-national study of this kind, focusing on a full-circle evaluation of the radiological capacity (actual equipment capacity, radiological protocols, human resources, experience of the available human resources, and the usage of the radiological equipment). The estimation of available radiological capacity and radiation dose exposure for children diagnosed and treated for TB will allow for an assessment of the existing gaps in radiological protocols and recommendations for subsequent optimisation.

Imaging techniques are useful and remain critical in the screening, diagnosis and monitoring of treatment responses in paediatric TB; however, in the current context of a revived call for the use of radiological images in paediatric TB by the WHO, it is equally critical for each country looking to expand the use of these techniques to evaluate their existing capacity and gaps. This aspect will be explored in our study and provide policy makers with some evidence which could be used to inform how imaging could be more optimally used in TB programs, particularly in low- and middle-income countries with a high TB burden, such as Mozambique and South Africa.

Despite the strength of our study, the design presents some limitations. First, the study will be based on routinely collected patient and programmatic data that are primarily designed for patient care and health system reporting and not obtained for research purposes. This may affect the quality of the data and introduce information bias. We will mitigate this by looking through all available paper-based and electronic medical records to reduce the amount of missing information as much as possible. The surveys given to clinicians and radiology staff are non-validated surveys and this will restrict the interpretation of the results. We will mitigate this by piloting the surveys to the respective professionals in other health facilities not involved in the study. The surveys will serve to complement the findings of the national- and patient-level data and, as such, are not the central aspect of the study. The characteristics of the selected population of children and the potential underestimation of radiation exposure (e.g., we did not include those diagnosed and not available for follow-up before treatment and those undiagnosed who may end up as fatalities) may restrict the generalisability of the study.

There will be recall bias and incomplete data in the measurement of frequency of imaging examinations in a TB episode. We extrapolate this measurement from different sources in the three countries to arrive at a rough estimate which may overestimate or underestimate our dosimetric findings. In this regard, caution should be exercised when analysing the findings. Selection bias in selecting study sites is an additional limitation. The study team have justified the reasons behind the site selections in each country and in the analysis will adjust for factors that can be controlled to break the biasing path.

Radiological procedures play a critical role in TB disease diagnosis and management in children, and their benefit is well documented and unquestionable. Whereas X-ray exposure from a single procedure is typically low, repeated examinations can lead to cumulative doses which might be of the order of a few tens of mGy to specific organs. As shown in the recent CT scan studies notably the large International Paediatric CT Scan Study (EPI-CT), there is a clear association between risk of brain cancers and haematological malignancies and doses typically received in CT scanning [34, 35]. It is therefore important to re-emphasize that the exposure of paediatric patients should be kept as low as reasonably achievable by appropriate justification of procedures and optimization of parameters. Our study aims to capture differences in exposure levels due to different health care practices (for example, the use of CT scans in Spain for the diagnosis of childhood TB, variations in radiological protocols), allowing for a more comprehensive understanding of the radiological framework in paediatric TB diagnosis internationally.

The planned comparative analysis will inform policy-makers, clinicians and radiologists on best practices and means to optimise the use of radiological technology in the management of paediatric TB. The results of our study will contribute to the identification of potential gaps in the capacity of radiological imaging techniques and radiological procedures. Potential optimisation of radiological protocols and policy recommendations will be developed from this process. Ultimately, the findings will assist in prioritising investment in TB programs. Furthermore, we anticipate that the findings of our research will also be useful for training and education purposes, identifying potential gaps in radiation protection optimization and recommending ways to keep health professionals informed about the latest developments in medical radiation.

## Acknowledgments

We would like to recognise the hard work and valuable contributions of all colleagues and partners in this project. We want to specifically mention: Lourdes Arjona, Alex Albert, Loide Cossa, Quique Bassat, James Seddon, Mqondisi Maphophe and Chris Buck. We also thank the national health authorities participating in this study (CSN, NTP Mozambique, Western Cape Department of Health and SAHPRA) as well as the participating study sites and patients in each of the countries. We would also like to acknowledge Global English Editing for the proof editing services.

## Author Contributions

**Conceptualization:** Isabelle Munyangaju, Richard D. Pitcher, Elisa López-Varela, Isabelle Thierry-Chef.

**Data curation:** Isabelle Munyangaju, Benedita José, Ridwaan Esmail, Megan Palmer, Begoña Santiago, Alicia Hernanz-Lobo, Crimenia Mutemba, Patricia Perez, Liebe Hendrietta Tlhapi, Vanessa Mudaly, Richard D. Pitcher, Andreas Jahnen, Eliseo Vañó Carruana.

**Formal analysis:** Isabelle Munyangaju, Benedita José, Elisa López-Varela, Isabelle Thierry-Chef.

**Funding acquisition:** Isabelle Thierry-Chef.

**Investigation:** Isabelle Munyangaju, Benedita José, Ridwaan Esmail, Megan Palmer, Begoña Santiago, Alicia Hernanz-Lobo, Eliseo Vañó Carruana.

**Methodology:** Isabelle Munyangaju, Isabelle Thierry-Chef.

**Project administration:** Isabelle Munyangaju.

**Resources:** Isabelle Thierry-Chef.

**Software:** Andreas Jahnen.

**Supervision:** Elisa López-Varela, Isabelle Thierry-Chef.

**Validation:** Elisa López-Varela, Isabelle Thierry-Chef.

**Visualization:** Richard D. Pitcher, Andreas Jahnen, Eliseo Vañó Carruana, Elisa López-Varela, Isabelle Thierry-Chef.

**Writing – original draft:** Isabelle Munyangaju, Benedita José, Ridwaan Esmail, Megan Palmer, Begoña Santiago, Alicia Hernanz-Lobo, Crimenia Mutemba, Patricia Perez, Liebe Hendrietta Tlhapi, Vanessa Mudaly, Richard D. Pitcher, Andreas Jahnen.

**Writing – review & editing:** Isabelle Munyangaju, Benedita José, Ridwaan Esmail, Megan Palmer, Begoña Santiago, Alicia Hernanz-Lobo, Crimenia Mutemba, Patricia Perez, Liebe Hendrietta Tlhapi, Vanessa Mudaly, Richard D. Pitcher, Andreas Jahnen, Eliseo Vañó Carruana, Elisa López-Varela, Isabelle Thierry-Chef.

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
