## [Decision Letter · Decision Letter 0]

10 Jan 2024

PONE-D-23-18845Evaluation of radiological capacity and usage in paediatric TB diagnosis: Protocol of a comparative study in Mozambique, South Africa and SpainPLOS ONE

Dear Dr. Munyangaju,

Thank you for submitting your manuscript to PLOS ONE. After careful consideration, we feel that it has merit but does not fully meet PLOS ONE’s publication criteria as it currently stands. Therefore, we invite you to submit a revised version of the manuscript that addresses the points raised during the review process.

**ACADEMIC EDITOR: Please insert comments here and delete this placeholder text when finished.** 

Please take your time to read and respond to the reviewers comments. In particular it is important to discuss potential limitations of this project.

We look forward to receiving your revised manuscript.

Kind regards,

Francis John Gilchrist

Academic Editor

PLOS ONE

Journal Requirements:

“We acknowledge support from the grant CEX2018-000806-S funded by MCIN/AEI/ 10.13039/501100011033, and support from the Generalitat de Catalunya through the CERCA Program. Isabelle Munyangaju the support of a fellowship from “la Caixa” Foundation (ID 100010434). The fellowship code is LCF/BQ/DI21/11860045. Role of the funding sources: None of the funding sources had a role in the study’s design, conduct and reporting.”

Additional Editor Comments (if provided):

Please take your time to read and respond to the reviewers comments. In particular it is important to discuss potential limitations of this project.

Reviewers' comments:

Reviewer's Responses to Questions

**Comments to the Author**

1. Does the manuscript provide a valid rationale for the proposed study, with clearly identified and justified research questions?

Reviewer #1: Yes

Reviewer #2: Yes

2. Is the protocol technically sound and planned in a manner that will lead to a meaningful outcome and allow testing the stated hypotheses?

Reviewer #1: Partly

Reviewer #2: Yes

3. Is the methodology feasible and described in sufficient detail to allow the work to be replicable?

Reviewer #1: Yes

Reviewer #2: Yes

4. Have the authors described where all data underlying the findings will be made available when the study is complete?

Reviewer #1: Yes

Reviewer #2: No

5. Is the manuscript presented in an intelligible fashion and written in standard English?

Reviewer #1: Yes

Reviewer #2: Yes

6. Review Comments to the Author

You may also provide optional suggestions and comments to authors that they might find helpful in planning their study.

Reviewer #1: The proposed study is interesting and ambitious; it will likely require a significant amount of effort to complete, collate and then to extract accurate findings. Obviously estimates of access to care will be major challenges in Mozambique and South Africa.

A few key comments:

1. First author has superscript (17) but no corresponding institution (17) is listed.

2. The "individual" level is better described as the "institutional" level. "Multi-country" is probably better described as "multi-national".

3. The use of "clinical and radiological human resources" is vague; e.g. compare line 176 with line 204. It seems that clinical medical physicists working in radiology are excluded entirely from this study and this may impact on the integrity of the data collection and analysis to estimate "cumulative radiation dose exposure".

4. Specific Objective No. 1: surely this is "...in Mozambique, South Africa and Spain by mapping the number and distribution of CXR units...."?

5. Clarification is required on Specific Objective No. 3. The inclusion (or not) of South African data does not consistent/clear in the manuscript, and is best described in lines 223 - 230. South Africa should be mentioned in Table 1 under this objective.

6. The study also does not seem to handle dosimetry across the sites in a consistent way, which means that data analysis could be become very complex, i.e. use of NCI software is only possible for 2 institutions. It may be better to rather use dose indicators or surrogates like image size, mAs and kVp for CXR, and mAS, scan length, scan thickness and pitch for CT, for example. Sub-analysis by age (number of exams and repeat imaging of individuals with more than one contact/episode) will be necessary if all patients under the age of 16y are to be included.

7. Line 178: please define, clarify or remove the phrase "radiological diagnostics".

8. The term "radiation protection (awareness)" is very broad, as this could imply that elements like radiation shielding, e.g. line 178. Consider using another term like optimization.

9. For patient records extracted over a number of years (Mozambique and Spain) one assumes that repeat episodes in the patient once they are over the age of 16y, are excluded from the study and will not count towards cumulative dose?

Reviewer #2: This protocol paper outlines a study that will address the capacity for radiological diagnosis of paediatric TB in low-and middle-income countries with a high TB burden. The methods appear rigorous. The strengths and anticipated limitations need to be stated more clearly.

Comments

1. Please make reference to the "mixed methods" design in the title for further clarification.

2. Please provide more up-to-date references 2023 in the background section i.e., Global TB report 2023 https://www.who.int/publications/i/item/9789240083851

3. Please clearly state the study outcomes for each objective in a table similar to Table 1 (or even in the same table).

4. Since adolescents are involved in the study, I think assent forms from those older than 12 or 13, depending on national regulations, are also necessary; however, the protocol makes no mention of this. Could you please elaborate?

5. Please include the current study status in the methods section.

6. Line 172: Could you please specify how many site operation managers are intended to receive the survey?

7. Please include a risk/benefit analysis in the protocol.

7. PLOS authors have the option to publish the peer review history of their article (what does this mean?). If published, this will include your full peer review and any attached files.

Reviewer #1: No

Reviewer #2: No

---

## [Author Response · Author response to Decision Letter 0]

18 Jan 2024

16th January 2023

Dear Editor,

We greatly appreciate the reviewers for their time and valuable comments, that have helped us improve the manuscript. Please find enclosed our point by point responses to them. The manuscript is attached in track changes to show the modifications and in a clean version.

Editor’s comments:

• The manuscript was modified as per the requirements of the journal

• IM, ELV and ITC acknowledge the support from the grant CEX2018-000806-S funded by MCIN/AEI/ 10.13039/501100011033, and support from the Generalitat de Catalunya through the CERCA Program. IM acknowledges the support of a fellowship from “la Caixa” Foundation (ID 100010434). The fellowship code is LCF/BQ/DI21/11860045. Role of the funding sources: None of the funding sources had a role in the study’s design, conduct and reporting. All other authors received no specific funding for this work. There was no additional external funding received for this study.

3. Data availability

• We adhere to the open data policy.

• We have deleted the ethics statement from the declaration section, it now only appears in methods.

5. Please include captions for your Supporting Information files at the end of your manuscript, and update any in-text citations to match accordingly. 

• We do not have supporting information associated with the manuscript.

Reviewer 1 comments:

1. First author has superscript (17) but no corresponding institution (17) is listed.

• This has been double checked, there are only 16 superscripts with corresponding institutions.

2. The "individual" level is better described as the "institutional" level. "Multi-country" is probably better described as "multi-national".

• Thank you. This has been corrected to “patient level” (to avoid confusion with hospital level) and “multi-national”.

3. The use of "clinical and radiological human resources" is vague; e.g. compare line 176 with line 204. It seems that clinical medical physicists working in radiology are excluded entirely from this study and this may impact on the integrity of the data collection and analysis to estimate "cumulative radiation dose exposure".

• To maintain comparability across the 3 countries, the study does not include medical physicists in the surveys. The sites selected in Mozambique have no medical physicists, and there are few medical physicists in the country. However, where possible, we had the support of medical physicists for the collection of radiological parameters and radiological protocols.

• This has been added in line 213-217 in the manuscript

4. Specific Objective No. 1: surely this is "...in Mozambique, South Africa and Spain by mapping the number and distribution of CXR units...."?

• Corrected as suggested

5. Clarification is required on Specific Objective No. 3. The inclusion (or not) of South African data does not consistent/clear in the manuscript, and is best described in lines 223 - 230. South Africa should be mentioned in Table 1 under this objective.

• Following recent discussion with partners in South Africa, we had the possibility to revise our strategy since we submitted the manuscript for review. An amendment has been submitted to the ethics review board in South Africa so we can align with local regulations and use the most accurate methodology (providing data which will be more comparable with those obtained in the other 2 countries). Radiological parameters for each TB case treated in year 2022 will be extracted, for each clinical record, by the site PACS manager. The associated clinical data will be extracted by site clinicians and each patient record will have a unique study number which will be provided to the study team. As a result, the study team will not be able to access patient identifiers, as required by the Protection of Personal Information Act.

• Corrected in line 232 – 239 as follows: In South Africa, this methodology will be adapted to align with local regulations and use the most accurate methodology (providing data which will be more comparable with those obtained in the other 2 countries). Radiological parameters for each TB case treated in year 2022 will be extracted, for each clinical record, by the site PACS manager. The associated clinical data will be extracted by site clinicians and each patient record will have a unique study number which will be provided to the study team. As a result, the study team will not be able to access patient identifiers, as required by the Protection of Personal Information Act.

6. The study also does not seem to handle dosimetry across the sites in a consistent way, which means that data analysis could be become very complex, i.e. use of NCI software is only possible for 2 institutions. It may be better to rather use dose indicators or surrogates like image size, mAs and kVp for CXR, and mAS, scan length, scan thickness and pitch for CT, for example. Sub-analysis by age (number of exams and repeat imaging of individuals with more than one contact/episode) will be necessary if all patients under the age of 16y are to be included.

• In all study sites, we will be able to collect the radiological parameters needed to use the NCI software. In South Africa and Spain, all parameters will be extracted from DICOM headers in PACS since this is available in all selected sites. The parameters will be extracted from DICOM images and available radiological protocols at the site in Mozambique.

In light of the reviewer's suggestion, we will carry out a sub-analysis based on age as well as analyse dose indicators in addition to organ doses.

7. Line 178: please define, clarify or remove the phrase "radiological diagnostics".

• This has been removed as suggested.

8. The term "radiation protection (awareness)" is very broad, as this could imply that elements like radiation shielding, e.g. line 178. Consider using another term like optimization.

• Thank you, we have replaced it with “radiation protection optimization”.

9. For patient records extracted over a number of years (Mozambique and Spain) one assumes that repeat episodes in the patient once they are over the age of 16y, are excluded from the study and will not count towards cumulative dose?

• This is correct. This study examines paediatric exposure, which excludes all exposures beyond 16 years of age.

Reviewer 2 comments:

1. Please make reference to the "mixed methods" design in the title for further clarification.

• Change made in the title as suggested: “Evaluation of radiological capacity and usage in paediatric TB diagnosis: a mixed-method protocol of a comparative study in Mozambique, South Africa and Spain”.

2. Please provide more up-to-date references 2023 in the background section i.e., Global TB report 2023 https://www.who.int/publications/i/item/9789240083851

• Thank you, the updates have been made.

3. Please clearly state the study outcomes for each objective in a table similar to Table 1 (or even in the same table).

• Thank you, we have added the study outcomes in Table 1 in line 137

4. Since adolescents are involved in the study, I think assent forms from those older than 12 or 13, depending on national regulations, are also necessary; however, the protocol makes no mention of this. Could you please elaborate?

• This has been included in line 229 -230 and 335

• Assent forms were included for Mozambique and Spain where a sample of legal guardians will be contacted to inquire about the number of CXR and/or CT scan the child had during a TB episode. 

• For all other surveys – only consent forms were included as the target population are adult healthcare professionals.

5. Please include the current study status in the methods section.

• This has been included in line 145: Currently, the study is in the data collection stage

6. Line 172: Could you please specify how many site operation managers are intended to receive the survey?

• This has been clarified. It is one operational site manager per site.

7. Please include a risk/benefit analysis in the protocol.

• Thank you very much for your comment and suggestion, we have added the following paragraph in the discussion line 386 – 399 (with additional references):

• Radiological procedures play a critical role in TB disease diagnosis and management in children, and their benefit is well documented and unquestionable. Whereas X-ray exposure from a single procedure is typically low, repeated examinations can lead to cumulative doses which might be of the order of a few tens of mGy to specific organs. As shown in the recent CT scan studies notably the large International Paediatric CT Scan Study (EPI-CT), there is a clear association between risk of brain cancers and haematological malignancies and doses typically received in CT scanning. It is therefore important to re-emphasize that the exposure of paediatric patients should be kept as low as reasonably achievable by appropriate justification of procedures and optimization of parameters. Our study aims to capture differences in exposure levels due to different health care practices (for example, the use of CT scans in Spain for the diagnosis of childhood TB, variations in radiological protocols), allowing for a more comprehensive understanding of the radiological framework in paediatric TB diagnosis internationally.

We hope that we have addressed their points satisfactorily and appreciate the opportunity to submit the new manuscript. All authors have seen and approved the revised version. We look forward to hearing from you soon.

Sincerely,

Isabelle Thierry-Chef

---

## [Decision Letter · Decision Letter 1]

8 Feb 2024

Evaluation of radiological capacity and usage in paediatric TB diagnosis: a mixed-method protocol of a comparative study in Mozambique, South Africa and Spain

PONE-D-23-18845R1

Dear Dr. Munyangaju,

We’re pleased to inform you that your manuscript has been judged scientifically suitable for publication and will be formally accepted for publication once it meets all outstanding technical requirements.

Kind regards,

Fran

Francis John Gilchrist

Academic Editor

PLOS ONE

Additional Editor Comments (optional):

Reviewers' comments:

Reviewer's Responses to Questions

**Comments to the Author**

1. Does the manuscript provide a valid rationale for the proposed study, with clearly identified and justified research questions?

Reviewer #1: Yes

2. Is the protocol technically sound and planned in a manner that will lead to a meaningful outcome and allow testing the stated hypotheses?

Reviewer #1: Yes

3. Is the methodology feasible and described in sufficient detail to allow the work to be replicable?

Reviewer #1: Yes

4. Have the authors described where all data underlying the findings will be made available when the study is complete?

Reviewer #1: Yes

5. Is the manuscript presented in an intelligible fashion and written in standard English?

Reviewer #1: Yes

6. Review Comments to the Author

You may also provide optional suggestions and comments to authors that they might find helpful in planning their study.

Reviewer #1: In the track changes version, line 172 has a repeat in the word "protection".

Thank you for the due consideration and implementation of the suggested changes.

7. PLOS authors have the option to publish the peer review history of their article (what does this mean?). If published, this will include your full peer review and any attached files.

Reviewer #1: No

---

## [Editor Report · Acceptance letter]

25 Mar 2024

PONE-D-23-18845R1 

PLOS ONE

Dear Dr. Munyangaju, 

I'm pleased to inform you that your manuscript has been deemed suitable for publication in PLOS ONE. Congratulations! Your manuscript is now being handed over to our production team.

Kind regards, 

on behalf of

Dr. Francis John Gilchrist 

Academic Editor

PLOS ONE